# Chrysin-Induced G Protein-Coupled Estrogen Receptor Activation Suppresses Pancreatic Cancer

**DOI:** 10.3390/ijms23179673

**Published:** 2022-08-26

**Authors:** Hyun Kyung Lim, Hee Jung Kwon, Ga Seul Lee, Jeong Hee Moon, Joohee Jung

**Affiliations:** 1Duksung Innovative Drug Center, Duksung Women’s University, Seoul 01369, Korea; 2College of Pharmacy, Duksung Women’s University, Seoul 01369, Korea; 3Disease Target Structure Research Center, Korea Research Institute of Bioscience & Biotechnology, Daejeon 34141, Korea; 4College of Pharmacy, Chungbuk National University, Cheongju 28644, Korea

**Keywords:** pancreatic cancer, chrysin, GPER, anticancer effect

## Abstract

Pancreatic cancer (PC) has a high mortality rate due to its poor prognosis and the possibility of surgical resection in patients with the disease. Importantly, adjuvant chemotherapy is necessary to improve PC prognosis. Chrysin, a natural product with anti-inflammatory, antioxidant, and anticancer properties, has been studied for several years. Our previous study demonstrated that chrysin induced G protein-coupled estrogen receptor (GPER) expression and regulated its activity in breast cancer. Herein, we investigated whether chrysin-induced GPER activation suppresses PC progression in MIA PaCa-2 cells and a xenograft model. To determine its mechanism of action, cytotoxicity and clonogenic assays, a FACS analysis, and Western blotting were performed. Furthermore, the delay in tumor growth was evaluated in the MIA PaCa-2-derived xenograft model. Tumor tissues were investigated by Western blotting, immunohistochemistry, and a proteomic analysis. Chrysin caused cell cycle arrest and significantly decreased cell viability. Following co-treatment with chrysin and 17β-estradiol, the inhibitory effect of chrysin on cell proliferation was enhanced. In the xenograft model, chrysin and G1 (a GPER agonist) significantly delayed tumor growth and reduced both Ki-67 (a proliferation marker) and c-Myc expressions in tumor tissues. The proteomic analysis of tumor tissues identified that rho-associated coiled-coil containing protein kinase 1 (ROCK1), transgelin 2 (TAGLN2), and FCH and Mu domain containing endocytic adaptor 2 (FCHO2) levels were significantly reduced in chrysin-treated tumor tissues. High *ROCK1*, *TAGLN2*, and *FCHO2* expressions were indicative of low overall PC survival as found using the Kaplan–Meier plotter. In conclusion, our results suggest that chrysin suppresses PC progression through the activation of GPER and reductions in ROCK1, TAGLN2, and FCHO2 expressions.

## 1. Introduction

Pancreatic cancer (PC) is the seventh leading cause of cancer-related deaths worldwide [1]. Although its exact cause is unknown, it is associated with smoking, obesity, genetics, and diabetes [2]. As there are often no apparent symptoms in the early stages of the disease, it is often only detected when the disease has already significantly progressed. As such, PC has a poor prognosis, with a 5-year survival rate between 11% and 12%, which is notably low when compared to that of other cancers [3,4]. PC is typically treated using surgical interventions and chemotherapy. Surgery is the most effective treatment option, although only approximately 20% of patients with PC are diagnosed with the surgically resectable type [5]. Chemotherapy involves the use of 5-fluorouracil (5-FU) or gemcitabine, which both have several side effects [6]. Furthermore, owing to the difficulties of drug penetration as a result of pancreatic fibrosis, PC has low drug responsiveness [7]. Consequently, to enhance the overall survival of patients with PC, effective anticancer treatments are required.

Estrogen, a sex hormone, acts as a ligand for estrogen receptors (ERs). Estrogen-activated ERs regulate the cell cycle, apoptotic proteins, and the proliferation of normal and cancerous cells [8,9]. It has been reported that estrogen and ERs are associated with non-reproductive cancers and diseases, such as colon cancer, osteoporosis, and neurodegenerative diseases, as well as reproductive system carcinomas, including those of the breast, prostate, and endometrium [10]. In one study, when PC cells were treated with 17β-estradiol (E2), the nanomolar concentrations of E2 stimulated cell proliferation by 40%, whilst the micromolar concentrations of E2 impaired PC proliferation [11]. In addition, phytoestrogens have been reported to inhibit or stimulate the proliferation of PC [12], suggesting that estrogen may affect disease progression. Several epidemiological studies have reported that estrogen-related hormonal therapy reduces the risk of pancreatic cancer [13,14].

ER subtypes, activated by estrogen, include ER-α, ER-β, and the G protein-coupled estrogen receptor (GPER). GPER, which contains seven transmembrane domains, mediates non-genomic estrogenic effects by acting specifically on estrogen and related analogs [15]. Furthermore, it is involved in nervous, immune, cardiovascular, and reproductive systems, as well as in bone metabolism, pancreatic glucose metabolism, and renal function [16]. Ligands, such as estrogen, activate GPER, in turn activating subtype units alpha, beta, and gamma. In addition, the stimulation of GPER activates cell signaling genes, such as MAPK and c-Myc (proliferative and growth-stimulating transcription factor regulators, respectively) [17,18]. GPER has been implicated in various malignancies, such as hepatocellular, breast, and colorectal cancers [19,20,21].

Chrysin (5,7-dihydroxy-2-phenyl-4H-chromen-4-one) is a natural flavonoid found in propolis, honey, and various plants [22]. Chrysin is a phytoestrogen that can act as a ligand for ERs [23], and it exhibits anti-inflammatory, anti-viral, and anti-aging properties, as well as therapeutic effects against AIDS, diabetes, and cancer [24]. In previous studies, chrysin was reported to enhance the therapeutic efficacy of the chemotherapy agent docetaxel, resulting in an enhanced inhibition of cell proliferation in non-small-cell lung cancer A549 cells [25,26]. Additionally, we have previously demonstrated that chrysin induces GPER expression and regulates GPER activity in breast cancer [27].

In this study, we investigated the action and molecular mechanism of chrysin on PC progression using MIA PaCa-2 cells and a xenograft model.

## 2. Results

### 2.1. Cytotoxic Effect of Chrysin in MIA PaCa-2 Cells

The anticancer effects of chrysin were evaluated in MIA PaCa-2 cells (Figure 1A). Chrysin significantly inhibited cell viability in a dose- and time-dependent manner. To investigate chrysin-induced apoptosis, the expression levels of caspase-9, procaspase-3, and cleaved PARP were determined in chrysin-treated MIA PaCa-2 cells (Figure 1B). We elucidated that chrysin cleaved and activated caspase-9, and then it activated caspase-3. Subsequently, chrysin induced PARP cleavage. These results indicate that chrysin induced endogenous apoptosis, resulting in the inhibition of cell viability.

### 2.2. Effect of E2 on Chrysin-Related Inhibition of MIA PaCa-2 Cell Proliferation

The expression levels of ERs were investigated to elucidate the association between E2 and PC progression (Figure 2A). Chrysin only increased GPER expression levels, as observed in our previous study [27]. E2 (10 nM) did not change the expression of ERs, whilst the combination of chrysin and E2 slightly decreased ERα expression and increased the expression of GPER. In the MTT assay, chrysin-induced cytotoxicity did not change with the addition of E2 (Figure 2B). However, the combination of chrysin and E2 potentiated the inhibition of cell proliferation by chrysin, whereas E2 alone did not inhibit cell proliferation (Figure 2C). We investigated whether these observed results of the clonogenic assay were caused by cell cycle arrest (Figure 2D). The control and E2 showed the same effects on the cell cycle; however, chrysin, and the combination of chrysin and E2, arrested the G2/M phase. In addition, we investigated c-Myc expression levels, as c-Myc is associated with GPER activation and cell proliferation. Chrysin decreased c-Myc expression, and the combination of chrysin and E2 enhanced this decrease in expression (Figure 2E).

### 2.3. Inhibition of Cell Proliferation Using a GPER Agonist

To elucidate the relevance of GPER in the progression of PC, G1 was used as a GPER agonist. G1 and chrysin induced GPER expression (Figure 3A). Furthermore, G1 inhibited cell viability (Figure 3B). As depicted in Figure 3C, G1 inhibited cell proliferation, whereas G15, a GPER antagonist, had no effect on cell proliferation. The combination of chrysin and G1 also inhibited colony formation, although chrysin itself was found to be more potent, indicating that chrysin and G1 could competitively bind to GPER. The combination of chrysin and G1 inhibited the formation of colonies less than chrysin alone, suggesting that G15 caused partial antagonism of its target receptors.

### 2.4. Chrysin-Related Tumor Growth Delay in a MIA PaCa-2-Cell-Derived Xenograft Model

In in vitro assays, chrysin showed an anticancer effect via GPER targeting, as shown in Figure 1, Figure 2 and Figure 3. Therefore, we evaluated the anticancer effects of chrysin in a xenograft model. Chrysin (50 mg/kg, p.o.) and G1 (10 mg/kg, i.p.) treatments significantly delayed tumor growth (Figure 4A). To investigate physiological changes in tumor tissues, Ki-67 (a proliferation marker) and c-Myc expression levels were determined using immunohistochemistry. As shown in Figure 4B,C, the expressions of Ki-67 and c-Myc were inhibited in chrysin- and G1-treated tumor tissues. These results are consistent with the previously discussed results obtained in vitro.

### 2.5. Chrysin-Associated Molecular Factors in the Inhibition of Pancreatic Cancer Progression

To identify the molecular factors in tumor tissues, a proteomic analysis was performed. Among the 3188 proteins tested, those that were differentially expressed between the control and chrysin-treated tumor tissues are shown in Table 1. The following factors were identified: serine and arginine repetitive matrix 1 (SRRM1), transgelin 2 (TAGLN2), FCH and Mu domain containing endocytic adaptor 2 (FCHO2), DENN domain containing 1A (DENND1A), zinc finger CCCH-type containing 18 (ZC3H18), cadherin 11 (CDH11), FAST kinase domains 2 (FASTKD2), and rho-associated coiled-coil containing protein kinase 1 (ROCK1). They were selected using the following criteria: fold change ≤0.7 and *p* value < 0.05. These genes are represented in red spots in Figure 5.

We investigated the association of these proteins with GPER using the STRING interaction network. As shown in Figure 6A, ROCK1, TAGLN2, FCHO2, and GPER showed an indirect interaction via epidermal growth factor receptor (EGFR). The correlation between these gene expressions and overall survival (OS) was investigated using the KM plotter (www.kmplot.com (accessed on 31 July 2022)). To analyze the data, we used the mRNA sequencing and OS of patients with pancreatic ductal adenocarcinoma (n = 177) and set the best cutoff to compare their OS with high and low gene expression (Figure 6B). In the case of GPER, we considered that estrogen could regulate the action of GPER, and then we compared the correlation between *GPER* expression and the OS in female and male patients with PC. A high *GPER* expression decreased the hazard ratio (HR) and delayed the OS of patients with PC. Interestingly, *GPER* levels were significantly positively correlated with OS in female patients with PC (*p* = 0.0086). Low *ROCK1*, *TAGLN2*, and *FCHO2* expressions delayed the OS of patients with PC. These results suggest that chrysin-induced GPER is associated with a decrease in ROCK1, TAGLN2, and FCHO2 levels, which may be involved in the OS of patients with PC.

## 3. Discussion

Various studies have investigated various therapeutic strategies to overcome the poor prognosis of PC [28]. Several studies identifying a lower risk of PC in premenopausal women suggest that the levels of sex hormones may be correlated with disease progression. However, the role of sex hormone receptors in PC remains controversial [29,30,31]. Additionally, the β/α ER expression ratio is regarded as the main factor in estrogen-related therapy for PC [32]. Our study findings suggest that chrysin-activated GPER could control disease progression by decreasing the expressions of ROCK1, TAGLN2, and FCHO2. These results support the hypothesis that sex hormones and their receptors are associated with the development and progression of PC.

Several studies have reported that flavonoids, such as fisetin and kaempferol, exhibit anticancer effects in PC cells. Previous results have demonstrated that, in PC cells, fisetin enhanced gemcitabine-induced cytotoxicity [33], whereas kaempferol induced apoptosis through the upregulation of reactive oxygen species [34]. Chrysin is a flavonoid with anti-inflammatory, antioxidant, and anticancer properties. In previous studies, chrysin was reported to induce G2/M phase arrest in colon cancer cells [35], activate p38 and NK-κB/p65 in cervical cancer HeLa cells [36], induce apoptosis in MDA-MB-231 cells [26], enhance docetaxel-induced cytotoxicity in A549 cells [25], induce ROS-dependent autophagy through binding and inhibiting human carbonyl reductase 1, and improve the sensitivity of gemcitabine in PC cells [37]. Our results elucidate the role of chrysin in PC using in vitro and in vivo models (Figure 2 and Figure 4). The results shown in Figure 1 and Figure 2 are consistent with those of previous studies. Interestingly, estrogen potentiated chrysin-induced cytotoxicity, suggesting that the induction of GPER expression might play an important role. Thus, G1, a GPER agonist, showed similar results to E2 (Figure 3). G15, a GPER antagonist, slightly inhibited chrysin-induced cytotoxicity. These results indicate that GPER regulation can control the progression of PC.

Consistent with our results, a high GPER expression was associated with prolonged OS in patients with PC (Figure 5). In particular, the OS differences in GPER expression were significant in women. To elucidate the mechanism of action of chrysin through GPER, the expression levels of several factors were investigated in tumor tissues. The c-Myc protein is known to be downregulated in normal cells but overexpressed in various cancers, and it is associated with cell proliferation and differentiation [38]. Thus, the inhibition of c-Myc in cancer therapy has been reported in various studies [39,40,41]. In melanoma, GPER has been reported to exert anticancer effects through the inhibition of c-Myc [42]. The expression of the Ki-67 protein, a marker of cell proliferation, was found to be higher in tumor tissues than in normal tissues. Hu et al. reported that the overexpression of the Ki-67 protein in PC tissues is correlated with pathological grading, lymphatic metastasis, and the patient’s clinical stage [43]. Our results show that the chrysin- and G1-treated groups had decreased Ki-67 and c-Myc levels following GPER activation, resulting in significantly delayed tumor growth as compared to the control (Figure 4).

In particular, the differentially expressed genes between the control and chrysin-treated tumor tissues were investigated, and eight genes were identified (Table 1). The interaction of these genes and GPER were investigated using STRING proteomics. Three factors were identified: ROCK1, TAGLN2, and FCHO2. ROCK1, TAGLN2, FCHO2, and GPER all showed interactions in common with EGFR. Chrysin was reported to inhibit EGFR in breast cancer stem cells [44]. We will further study the correlation of chrysin and EGFR in PC. Remarkably, differentially expressed genes with low expressions in patients with PC prolonged the OS (Figure 6). ROCK1 plays a role in the metastasis of cellular movement and the accumulation of extracellular matrix in cancer-associated fibroblasts (CAFs), thus demonstrating that it is an important signaling pathway in cancer progression. In PC, ROCK1 is highly expressed, and its inhibition decreased tumor cell growth and CAFs in a previous study [45]. Additionally, TAGLN2 also shows a higher expression in PC cells than in normal corresponding cells, as previously reported [46]. Furthermore, the knock-out of FCHO2 increased chemosensitivity in PC MIA-PaCa2 cells in another study [47].

In conclusion, our results suggest that chrysin-induced GPER activation decreases ROCK1, TAGLN2, and FCHO2 expressions and subsequently suppresses the proliferation of PC in a MIA PaCa-2-cellsderived xenograft model. Therefore, chrysin-like agents could have a role in PC therapy.

## 4. Materials and Methods

### 4.1. Samples

As shown in Figure 7, chrysin (Sigma-Aldrich, St. Louis, MO, USA), E2 (Sigma-Aldrich), G1, and G15 (Cayman Chemical, Ann Arbor, MI, USA) were used as GPER ligands.

### 4.2. Cell Culture

The human pancreatic cancer cell line MIA PaCa-2 was purchased from the Korean Cell Line Bank and cultured with 1% penicillin–streptomycin (GenDEPOT, Katy, TX, USA) and 10% fetal bovine serum (GW Vitek, Seoul, Korea) in Dulbecco’s modified Eagle’s medium (DMEM, GenDEPOT) at 37 °C in a 5% CO_2_ atmosphere.

### 4.3. Cell Viability Assay

MIA PaCa-2 cells were seeded in 96-well plates at a density of 1.0 × 10^4^ or 0.8 × 10^4^ cells/well. After 24 h, chrysin, E2, G1, or G15 were added to each well for 24 or 48 h. MTT (Sigma-Aldrich) solution (10 μL: 5 mg/mL in PBS) was then added to plates and incubated at 37 °C for 3 h. Next, the medium was discarded, and 100 μL DMSO (Sigma-Aldrich) was added prior to further incubation at 37 °C for 30 min in the dark. Absorbances were measured at 560 nm using a microplate reader (Infinite M200 PRO, TECAN, Grödig, Austria).

### 4.4. Clonogenic Assay

Cells were seeded in 6-well plates at densities of 100 (control), 500 (single treatment), and 1000 (combination treatment) cells/well. The cells treated with the sample were incubated, and colonies were formed. Once a colony composed of over 50 cells was observed in the control, all wells were stained with 0.5% crystal violet in 10% methanol for 15 min prior to being washed with PBS. The number of colonies was counted using AlphaView software (version 3.3.0, ProteinSimple, Santa Clara, CA, USA), and the survival fraction (SF) was calculated by correcting the plating efficiency (PE).
PE = (Number of colonies in the control group)/(Number of seeded cells in the control group)
SF = (Number of colonies in the treated group)/(Number of seeded cells in the treated group × PE)

### 4.5. Western Blot Analysis

Cells were seeded in a 6-well plate at a density of 2 × 10^5^ cells/well. After 24 h, the cells were incubated for 48 h. The cells were harvested and lysed in RIPA buffer (GenDEPOT) containing inhibitors (Xpert protease inhibitor and phosphatase inhibitor cocktail solution, GenDEPOT). Cell lysates were denatured using sample buffer and separated using 10% SDS-PAGE. Proteins were transferred onto PVDF membranes (Millipore, Darmstadt, Germany) using a semi-dry electroblotting apparatus (PeqLab, Darmstadt, Germany). Membranes were blocked with 5% skim milk in TBST containing 50 mM Tris-HCl (pH 7.4), 150 mM NaCl, and 0.1% Tween 20, and they were incubated sequentially with primary antibodies at 4 °C overnight. After washing the membranes with TBST, they were incubated with secondary antibodies at 25 °C for 2 h. Immunoreactive proteins were visualized using ECL reagents and detected using the ChemiDoc apparatus (FluorChem E system, San Jose, CA, USA). The antibodies and ratios used were as follows: cleaved-PARP, caspase 3, and caspase 9: 1:1000 (Cell Signaling Technology, Danvers, MA, USA); GPER: 1:1000 (Abcam, Waltham, MA, USA); ER-α: 1:500 (Santa Cruz Biotechnology, Dallas, TX, USA); ER-β: 1:1000 (Merck KGaA, Darmstadt, Germany); β-actin: 1:5000 (Sigma-Aldrich); and anti-mouse IgG (H + L) horseradish peroxidase conjugate and anti-rabbit IgG (H + L) horseradish peroxidase conjugate: 1:3000 (Bio-Rad Laboratories, Hercules, CA, USA).

### 4.6. Cell Cycle Analysis

Cells were seeded in a 100 mm dish at a density of 2.4 × 10^5^ cells/dish. The following day, cells were treated for 8 h with chrysin, E2, or chrysin + E2. After 8 h, the cells were fixed with 70% ethanol in PBS, and dead cells were stained with propidium iodide (PI). The cell cycle was examined using Novocyte2000 (Acea Biosciences, Inc., San Diego, CA, USA).

### 4.7. Tumor Growth

Male BALB/c-nude mice (5-weeks old) were purchased from JA BIO (Gyeonggi, Korea). After acclimatization for a week, human pancreatic cancer MIA PaCa-2 cells were subcutaneously implanted into the right flank of each animal. When the tumor volume reached approximately 50 mm^3^, the mice were randomized into three groups and treated as follows: G1: 10 mg/kg by intraperitoneal injection (i.p.), three days a week; chrysin: 50 mg/kg by oral (p.o.) administration, five days a week; or vehicle (p.o.) for 5 weeks. Tumor volume was measured using a vernier caliper three times per week and calculated using the following equation:Tumor volume = [Length × (width)^2^]/2

### 4.8. Immunohistochemistry

Tumor tissues isolated from MIA PaCa-2-cell-derived xenografts were embedded in an optimal cutting temperature compound and snap-frozen in isopropanol using liquid nitrogen. Tissue sections (5 μm thick) were obtained using a cryostat (Leica, Nussloch, Germany). Each slide section was rehydrated with tap water and incubated with 3% H_2_O_2_ in methanol to quench endogenous peroxidase activity. After sections were blocked with 10% BSA in PBST to reduce non-specific interactions, the samples were incubated with primary antibodies at 4 °C overnight. The tissue slides were washed with PBS and subsequently incubated with secondary and tertiary antibodies (ABC kit, Vector Laboratories, Newark, CA, USA). Next, the sections were stained with DAB solution, and the nuclei in the tissues were counterstained with hematoxylin (Sigma-Aldrich). Stained tissue was observed under a microscope and counted using Image J software (v1.53t, NIH).

### 4.9. Proteomics Data Acquisition

Tumor tissues treated with a radioimmunoprecipitation assay (RIPA) buffer (GenDEPOT) containing protease inhibitor (GenDEPOT) and phosphatase inhibitor (Roche, Basel, Switzerland) were homogenized on ice. The lysate was added to 10% SDS (final concentration, 2%) and ultra-sonicated (QSONICA) for 30 s (on/off 3 s) at an amplitude of 50 to break DNA/RNA. Following this, the lysate was heated at 95 °C for 10 min, and the protein extract was centrifuged at 13,000× *g* for 10 min at 25 °C. The proteins were digested using Strap mini (PROTIFI, C02-mini-80, Farmingdale, NY, USA) according to the manufacturer’s protocol. The eluted peptides were dried using a speed-vac and labeled with TMT 10 plex (Thermo Scientific, 90111, Waltham, MA, USA). The labeled peptides were mixed and desalted using an Oasis HLB sorbent (Waters, WAT186000383). The mixed peptides were dried and dissolved in 5% acetonitrile (ACN)/10 mM ammonium bicarbonate. The fractionated samples were separated using an Acquity UPLC system (Waters, Milford, MA, USA) equipped with a BEH C18 column (1.7 μm, 2.1 × 100 mm, PN 186002352) and a fraction collector (Gilson, FC203B, Middleton, WI, USA). The separation conditions were under gradient elution from 5% to 60% solution, and a flow rate of 0.2 mL/min was used for fractionation. Buffers A and B contained 10 mM ammonium bicarbonate in 90% ACN [48].

To identify peptides using the LC/MS/MS system, each sample was dissolved in 0.1% formic acid/5% ACN and loaded onto the trap column (Acclaim PepMap 100, 75 μm × 2 cm, C18, 3 μm, PN 164946) in an RSLCnano u3000/Orbitrap Exploris 240 (Thermo Scientific) system. The peptides were separated by a BEH300 C18 column (1.7 μm, 75 μm × 25 cm, PN 186003815) at 50 °C. The gradient mobile phases were from 5% to 60% ACN with 0.1% formic acid at a flow rate of 300 nL/min. The conditions of the survey scan were as follows: resolution = 120,000, Max IT = auto, AGC 300%, and mass range = 400–1600. The parameters for the MS/MS scan were as follows: Top15 double play, resolution = 45,000, max IT = 80 ms, threshold 2E4, normalized collision energy = 36%, isolation width = 0.7, dynamic exclusion parameter excluded after n times = 1, exclusion duration time = 45 s, and mass tolerance low/high = 10 ppm.

Raw LC/MS/MS data were analyzed using Maxquant v2.0.3.0 (Max-Planck-Institut, Für biochemie, Planegg, Germany) using the following settings: database = UniProt Homo sapiens, enzyme = trypsin/P, variable modification = oxidation (M), acetyl (protein N-term), fixed modification = methylthio (C), and type = reporter ion MS2 TMT10plex. Among the various result files, proteinGroups.txt was used for further statistical calculations using Perseus v1.5.8.0 (Max-Planck-Institut). Protein groups were selected with the constraint that each experimental group contained at least two valid values. Missing values were replaced by a normal distribution. Lastly, the *p* value was calculated for two of the three experimental groups, and significant protein groups were selected based on fold changes and *p* values.

### 4.10. Overall Survival (OS) Analysis of Differentially Expressed Genes (DEGs)

The OS of *GPER, ROCK1, TAGLN2*, and *FCHO2* expression data from patients with pancreatic ductal adenocarcinoma cancer were obtained from mRNA sequences in the Kaplan–Meier plotter database (http://kmplet.com/analysis/, accessed on 31 July 2022). The log-rank *p* value and hazard ratios (HRs) with 95% confidence intervals were calculated using the Kaplan–Meier plotter.

### 4.11. Protein-Protein Interaction

A search tool for the retrieval of interacting genes (STRING) was used to investigate protein–protein interactions [49]. Potential correlations between GPER, ROCK1, TAGLN2, and FCHO2 were examined.

### 4.12. Statistical Analysis

Data are expressed as mean ± standard deviation (SD). Datasets were analyzed using a one-way or two-way ANOVA test followed by post hoc Tukey’s or Dunnett’s multiple comparisons test using GraphPad Prism 7 (GraphPad Software Inc., San Diego, CA, USA). Statistical significance was set at *p* < 0.05.

## Figures and Tables

**Figure 1 ijms-23-09673-f001:**
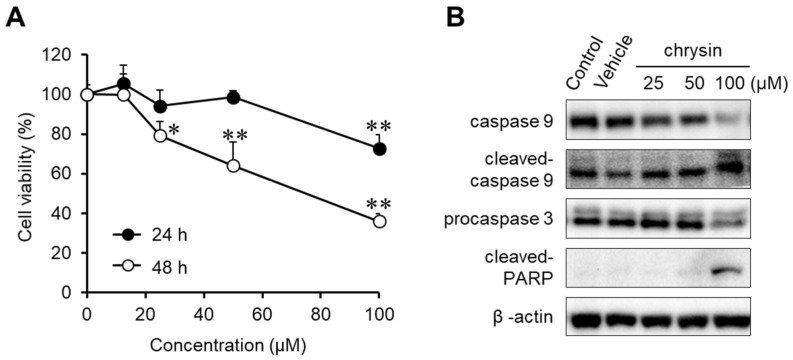
Anticancer effect of chrysin in MIA PaCa-2 cells. (**A**) Cytotoxicity of chrysin. Cell viability was determined by MTT assay. Data represent the mean ± standard deviation (n = 8). *, *p* < 0.01, **, *p* < 0.0001 (ANOVA one-way, Tukey’s post hoc test). (**B**) The expression of apoptosis-associated proteins by chrysin. Protein levels were measured by Western blotting analysis.

**Figure 2 ijms-23-09673-f002:**
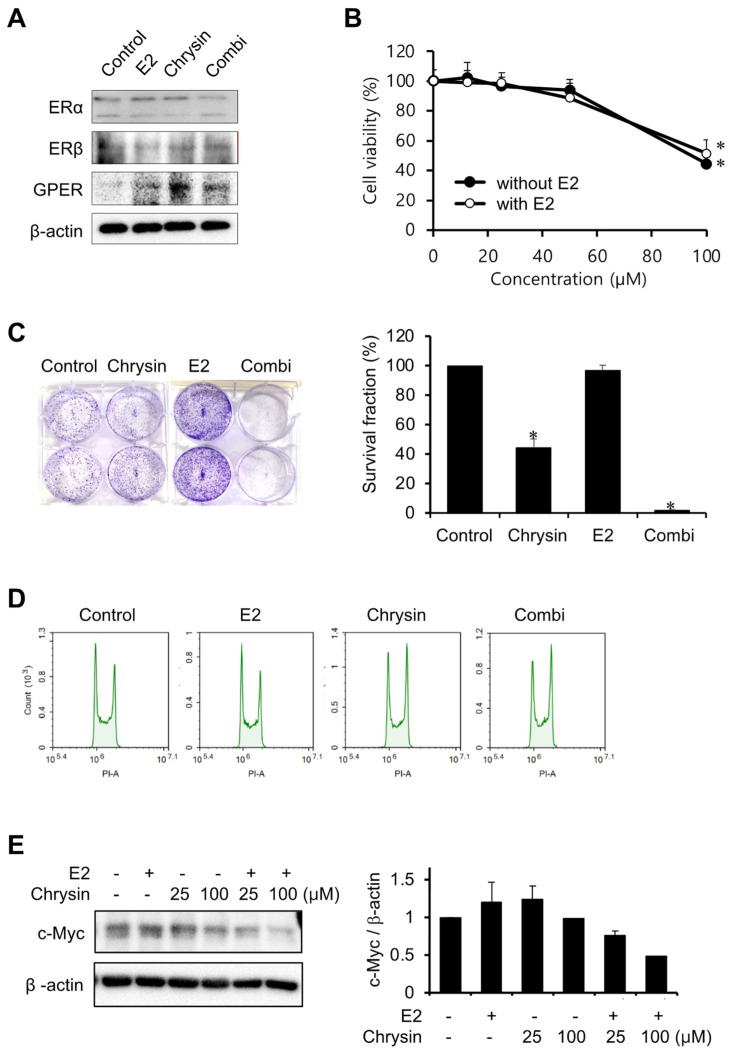
Potentiation of E2 on inhibition of cell proliferation by chrysin. (**A**) The expression of ERs. MIA PaCa-2 cells were treated with E2 (10 nM), chrysin (25 μM), or the combination of E2 and chrysin (Combi) for 48 h. (**B**) Cytotoxicity of chrysin with E2. MIA PaCa-2 cells were treated with various concentrations of chrysin with or without E2 (10 nM) for 48 h. Data represent the mean ± standard deviation (n = 7). *, *p* < 0.0001 (ANOVA one-way, Tukey’s post hoc test). (**C**) Inhibition of cell proliferation by chrysin. Photos of colonies following clonogenic assays (**left**). The graph shows the survival fraction (%), calculated as described in Materials and Methods (**right**). Data represent the mean ± standard deviation (n = 3). *, *p* < 0.0001 (ANOVA one-way, Tukey’s post hoc test). (**D**) Chrysin-induced cell cycle arrest. MIA PaCa-2 cells were treated with E2 (10 nM), chrysin (25 μM), or the combination of E2 and chrysin (Combi) for 8 h. (**E**) Decrease in c-Myc following chrysin treatment. The expression level of c-Myc was determined by Western blotting analysis and calculated using Image J (v1.53t, NIH).

**Figure 3 ijms-23-09673-f003:**
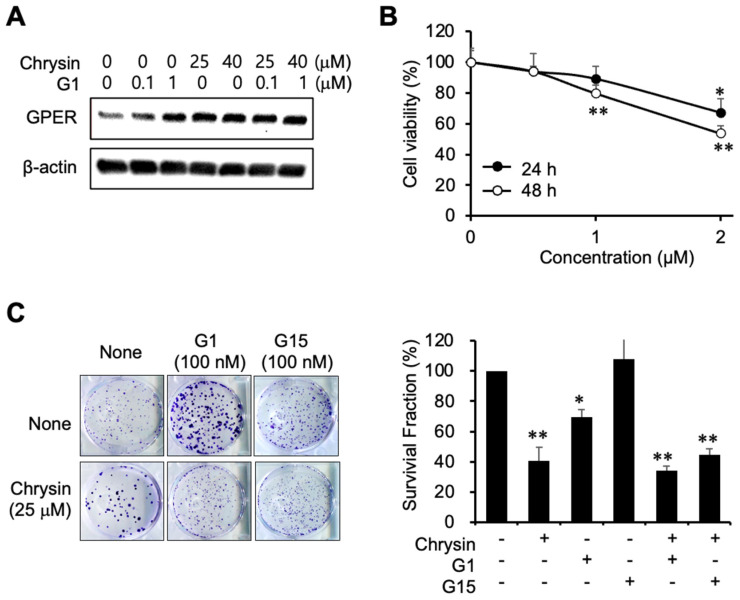
Inhibition of cell proliferation using a GPER agonist. (**A**) GPER expression following G1 and chrysin treatment. (**B**) Cytotoxicity of G1 in MIA PaCa-2 cells. Cell viability was determined by MTT assay. Data represent the mean ± standard deviation (n = 6). *, *p* < 0.001, **, *p* < 0.0001 (ANOVA one-way, Tukey’s post hoc test). (**C**) Inhibition of colony formation by G1 and chrysin in a clonogenic assay. Photos of colonies following the clonogenic assay (**left**). The graph shows the survival fraction (%), calculated as described in Materials and Methods (**right**). Data represent the mean ± standard deviation (n = 3). *, *p* < 0.001, **, *p* < 0.0001 (ANOVA one-way, Tukey’s post hoc test).

**Figure 4 ijms-23-09673-f004:**
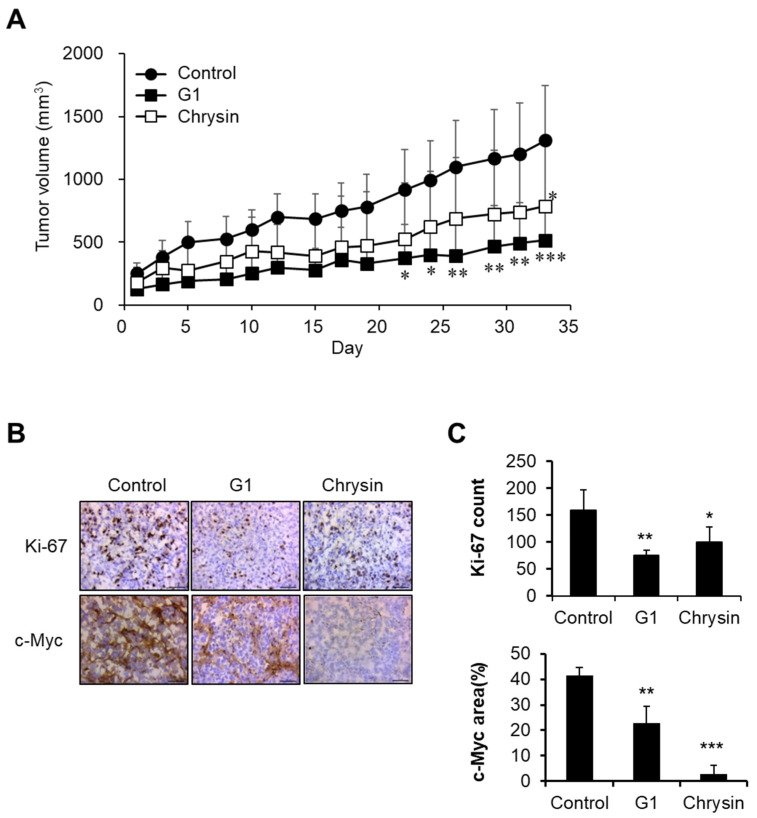
Inhibition of tumor growth by chrysin and G1. (**A**) Tumor growth delay following chrysin and G1 treatments. Data represent the mean ± standard deviation (n = 4). *, *p* < 0.05, **, *p* < 0.01, ***, *p* < 0.001 (ANOVA two-way, Dunnett’s post hoc test). (**B**,**C**) The expression levels of Ki-67 and c-Myc in tumor tissues. Photos were observed by microscopy (×40) (**B**), and data were calculated (**C**). Data represent the average ± standard deviation (Ki-67 count, n = 4, c-Myc area, n = 3). *, *p* < 0.05, **, *p* < 0.01, ***, *p* < 0.001 (ANOVA one-way, Tukey’s post hoc test).

**Figure 5 ijms-23-09673-f005:**
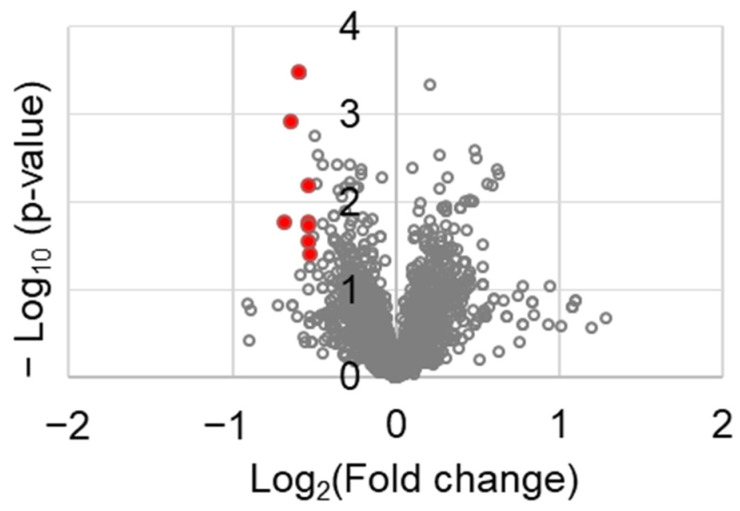
Volcano plot of the differentially expressed genes between control and chrysin-treated tumor tissues.

**Figure 6 ijms-23-09673-f006:**
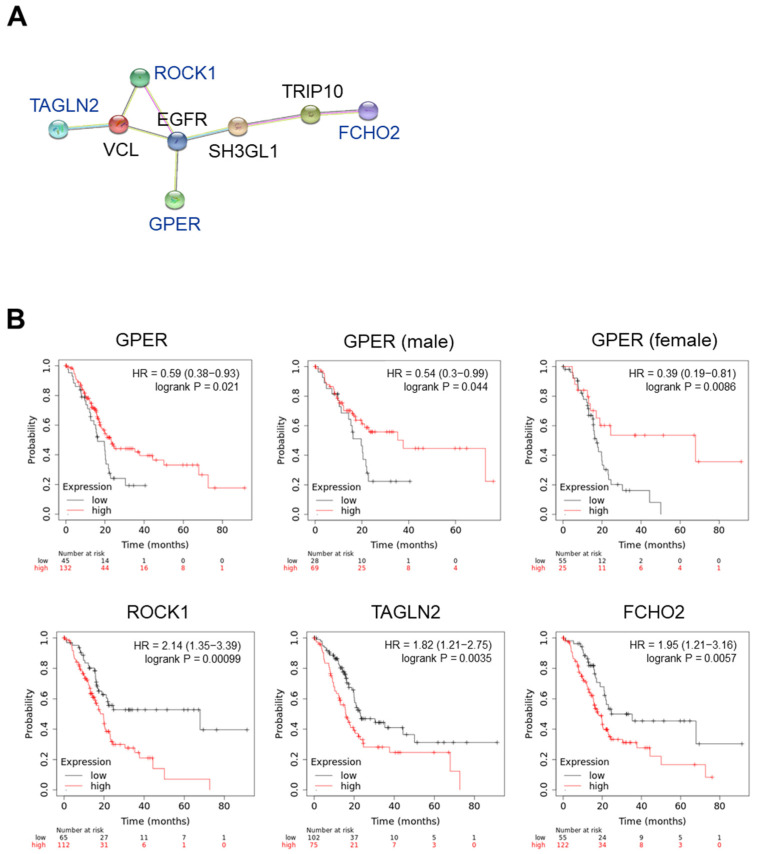
GPER-associated molecular factors in pancreatic cancer progression. (**A**) Protein–protein interaction networks in the STRING database. (**B**) Overall survival in patients with pancreatic cancer. Data were obtained following Kaplan–Meier analysis. HR, hazard ratio.

**Figure 7 ijms-23-09673-f007:**
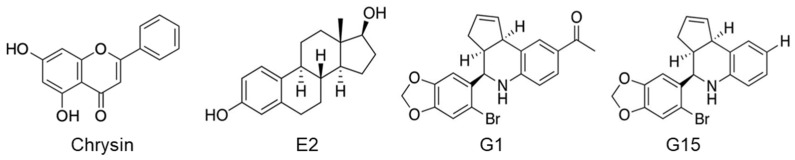
Structures of GPER ligands used in this study.

**Table 1 ijms-23-09673-t001:** Differentially expressed proteins between control and chrysin-treated tumor tissues.

Gene Symbol	Log (Fold Change)	*p*-Value
SRRM1	0.62	0.0170
TAGLN2	0.64	0.0012
FCHO2	0.67	0.0003
DENND1A	0.69	0.0173
ZC3H18	0.69	0.0065
CDH11	0.69	0.0191
FASTKD2	0.69	0.0289
ROCK1	0.70	0.0398

## Data Availability

All datasets generated for this study are included in this article.

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
