# Peer review of "Chrysin-Induced G Protein-Coupled Estrogen Receptor Activation Suppresses Pancreatic Cancer"

_ijms, 2022, doi:10.3390/ijms23179673_

Round 1
Reviewer 1 Report
In the present paper the authors report and comment the results obtained by their investigations aimed to provide further understanding on the pancreatic cancer that causes high mortality rate. The tests performed evidenced as particular proteins and receptors are targets of the natural product Chrysin used in the experiments.
The work is described in details for every method used.
Some issues must be addressed for the fact that the authors highlight as deeper molecular knowledge useful for elucidating the mechanism of action of chrysin can be obtained by their study, so at this purpose authors are invited to better clarify this aspect in the discussion. Furthermore more detailed conclusions are required.
Author Response
Reviewer#1
In the present paper the authors report and comment the results obtained by their investigations aimed to provide further understanding on the pancreatic cancer that causes high mortality rate. The tests performed evidenced as particular proteins and receptors are targets of the natural product Chrysin used in the experiments.
The work is described in details for every method used.
Some issues must be addressed for the fact that the authors highlight as deeper molecular knowledge useful for elucidating the mechanism of action of chrysin can be obtained by their study, so at this purpose authors are invited to better clarify this aspect in the discussion. Furthermore more detailed conclusions are required.
> Thank you for your valuable comment. To elucidate the mechanism of chrysin, our discussion is added more detail in this manuscript as you mentioned. (lines 240-245).
In the revised manuscript, we described the following:
Particularly, the differentially expressed genes between control and chrysin-treated tumor tissues were investigated, and found eight genes (Table 1). The interaction of these genes and GPER were investigated using STRING proteomics. Three factors were identified: ROCK1, TAGLN2, and FCHO2. ROCK1, TAGLN2, FCHO2, and GPER all showed interactions with EGFR in common. Chrysin was reported to inhibit EGFR in breast cancer stem cells [44]. We will further study the correlation of chrysin and EGFR in PC.
Reviewer 2 Report
Manuscript describes study on in vitro and in vivo investigation of chrysin as antiproliferative agent against pancreatic cancer. In general the manuscrip is well composed and could be of interest for the scientific community. Authors provided good evaluation on in vitro effects and short study in mouse model has been also described. There are however some issues that must be clarified.
First of all there are other studies on pancreatic cancers that utilize chrysin. The most important probably is work of Zhou 10.1016/j.bcp.2021.114813 that suggested mechanism involving ROS and ferroptosis. This work should be discussed (and cited).
I think that providing the structure of chrysin would be benefitial as well as providing just few more words about current status of chrysin and the contemporary research.
Name and reference for G1 agonst should be provided
References should be carefully corrected. There are many flaws - particularly in names of the journal source.
Author Response
Reviewer#2
Manuscript describes study on in vitro and in vivo investigation of chrysin as antiproliferative agent against pancreatic cancer. In general the manuscrip is well composed and could be of interest for the scientific community. Authors provided good evaluation on in vitro effects and short study in mouse model has been also described. There are however some issues that must be clarified.
First of all there are other studies on pancreatic cancers that utilize chrysin. The most important probably is work of Zhou 10.1016/j.bcp.2021.114813 that suggested mechanism involving ROS and ferroptosis. This work should be discussed (and cited).
>Thank you for your kind comment. The recent work of Zhou et al. is cited in this manuscript. (lines 217-219)
I think that providing the structure of chrysin would be benefitial as well as providing just few more words about current status of chrysin and the contemporary research.
>As you mentioned, we added Figure 6, which is the structures of chrysin, G1, G15, and estradiol. Furthermore, the current status of chrysin and the contemporary research were added. (lines 244)
Name and reference for G1 agonst should be provided
> As you mentioned, the name and reference for G1 were added in this manuscript. (lines 260-263)
References should be carefully corrected. There are many flaws - particularly in names of the journal source.
> Thank you for your helpful comment. We corrected the reference format according to “Instruction for Authors”.
Reviewer 3 Report
I am not able to recommend publication of this paper in its current form because I have some problems with the data and conclusions as presented. This is not to say that a revised paper would not be acceptable because there is a substantial body of data related to PC that could be important in finding a way to treat a very challenging disease, as the authors correctly point out. I ask the authors to consider the following:
1. The introduction to the paper ends with a simple statement of purpose (l 76) that does not match the conclusions. It would be helpful to the reader to give a little more information about the approach to be taken here.
2. I note the high concentration of chrysin required to produce measurable effects. No doubt this is associated with the unfavourable physicochemical properties of the compound but it matters because so many effects could be stimulated by chrysin. The contrast between the concentrations of the ligands A G1 and G15 required and that of chrysin is great and merits comment and evaluation.
3. There are some problems with the data. Many results do not reach statistical significance. Whilst not invalidating the results themselves, it does make the drawing of significant conclusions problematical.
4. There are some specific points on figure 2. Figure 2A is too grainy and small to evaluate objectively. Are there really two bands for ERalpha? In figure 2E, could the authors confirm that the units for chrysin are genuinely nanomolar because every other experiment shows micromolar, including figure 2A.
5. Is figure 3C really correct? The text states that G1 inhibits colony formation but the photograph of the G1 plate without chrysin (top row) appears to show many more colonies than in the control.
6. Unfortunately I find the data in figure 4A only indicative and not convincing because of the large uncertainties and only one data point at which chrysin showed a statistically significant result. I suspect that the concentration of chrysin reaching the xenograft was irregular and too low because of the route of administration and the intrinsically low activity of chrysin. A PK study is really needed to help design a more penetrating xenograft experiment.
7. In section 2.5 I could not find a Table 1. Also, what are the red points in figure 4A? My reading of the text is that the information for the Kaplan-Meier plots was taken from a data base which is not cited in the legend to figure 4 or elsewhere, as far as I could see. Because these data appear not to be original experimental data, they should be discussed with reference to a separate figure.
8. In reading the discussion, I find that the data as presented do not allow for the conclusions drawn to be made, in particular at lines 214 and 227. Moreover, there is no real evaluation of the data. No alternative hypotheses are presented. Consequently the impact of the study is very limited as presented.
I would like to emphasise that although I have been critical of the data and its presentation, it is possible that there is information of significance to the research community here. In the present form, however, I cannot recommend publication of this paper.
Author Response
Reviewer#3
I am not able to recommend publication of this paper in its current form because I have some problems with the data and conclusions as presented. This is not to say that a revised paper would not be acceptable because there is a substantial body of data related to PC that could be important in finding a way to treat a very challenging disease, as the authors correctly point out. I ask the authors to consider the following:
- The introduction to the paper ends with a simple statement of purpose (l 76) that does not match the conclusions. It would be helpful to the reader to give a little more information about the approach to be taken here.
> Thank you for your kind comments. We matched the sentence to the conclusion. We changed the following (lines 76-77):
In this study, we investigated the action and molecular mechanism of chrysin on PC progression using MIA PaCa-2 cells and a xenograft model.
- I note the high concentration of chrysin required to produce measurable effects. No doubt this is associated with the unfavourable physicochemical properties of the compound but it matters because so many effects could be stimulated by chrysin. The contrast between the concentrations of the ligands A G1 and G15 required and that of chrysin is great and merits comment and evaluation.
> I agree with your comment. The concentration of chrysin used in this study is higher than G1 and G15. Particularly, chrysin has various physiological properties. However, the complex effect of chrysin might avoid the inhibition of G15 (Fig. 3C).
- There are some problems with the data. Many results do not reach statistical significance. Whilst not invalidating the results themselves, it does make the drawing of significant conclusions problematical.
> Thank you for your comment. I agree with your comment. Several results were showed without significance; thus, we will need further study. Nevertheless, many results showed significant differences between the control and chrysin-treatment group. Therefore, a limited conclusion is suggested in the revised manuscript.
- There are some specific points on figure 2. Figure 2A is too grainy and small to evaluate objectively. Are there really two bands for ERalpha? In figure 2E, could the authors confirm that the units for chrysin are genuinely nanomolar because every other experiment shows micromolar, including figure 2A.
> I apologized for the incorrect writing. In this study, we treated micromoles of chrysin. We corrected that. We used an antibody against ERalpha obtained from SantaCruz (sc-130072). ER alpha isoforms are 66 kDa and 54 kDa, and ER46 is 48 kDa, ER36 is 36 kDa. We showed two bands of ER alpha.
Antibodies against ER alpha (sc-130072, sc-53493)
- Is figure 3C really correct? The text states that G1 inhibits colony formation but the photograph of the G1 plate without chrysin (top row) appears to show many more colonies than in the control.
> In the clonogenic assay, the seeding numbers of the control group and treated groups were different. The control group was 100 cells, but the treated groups were 500 cells because drug treatment-induced inhibition of colony formation was considered. In Figure 3C, the G1 plate seemed to be more colonies than the control, but the survival fraction is low, as shown in the right graph of Figure 3C.
We described in Materials and Methods the following (lines 279-280):
Cells were seeded in 6-well plates at densities of 100 (control), 500 (single treatment), and 1,000 (combination treatment) cells/well.
- Unfortunately I find the data in figure 4A only indicative and not convincing because of the large uncertainties and only one data point at which chrysin showed a statistically significant result. I suspect that the concentration of chrysin reaching the xenograft was irregular and too low because of the route of administration and the intrinsically low activity of chrysin. A PK study is really needed to help design a more penetrating xenograft experiment.
> I agreed with your comments. In the PK study of chrysin, chrysin (20 mg/kg or 100 mg/kg) was orally treated (1, 2). In a previous study (3), we also treated the same dose of chrysin in vivo model. Suppression of tumor growth in chrysin-treated groups was not significant except at the final date because of an individual difference. Nevertheless, tumor tissues of chrysin-treated groups showed anticancer effect (Fig. 4).
(1) Noh K, Oh do G, Nepal MR, Jeong KS, Choi Y, Kang MJ, Kang W, Jeong HG, Jeong TC. Pharmacokinetic Interaction of Chrysin with Caffeine in Rats. Biomol Ther (Seoul). 2016 Jul 1;24(4):446-52. doi: 10.4062/biomolther.2015.197. Epub 2016 Apr 25. PMID: 27098862; PMCID: PMC4930290.
(2) Shufan Ge, Song Gao, Taijun Yin, and Ming Hu, Determination of Pharmacokinetics of Chrysin and Its Conjugates in Wild-Type FVB and Bcrp1 Knockout Mice Using a Validated LC-MS/MS Method, Journal of Agricultural and Food Chemistry 2015 63 (11), 2902-2910. DOI: 10.1021/jf5056979
(3) Kim KM, Lim HK, Shim SH, Jung J. Improved chemotherapeutic efficacy of injectable chrysin encapsulated by copolymer nanoparticles. Int J Nanomedicine. 2017 Mar 9;12:1917-1925. doi: 10.2147/IJN.S132043. PMID: 28331315; PMCID: PMC5352247.
- In section 2.5 I could not find a Table 1. Also, what are the red points in figure 4A? My reading of the text is that the information for the Kaplan-Meier plots was taken from a data base which is not cited in the legend to figure 4 or elsewhere, as far as I could see. Because these data appear not to be original experimental data, they should be discussed with reference to a separate figure.
> Sorry for the inconvenience of the review. When this manuscript was edited for the format of this journal, Table 1 was omitted. In Figure 5, red points represented the differentially expressed genes (Table 1). Omitted information (red points and KM plot database) was added. As you mentioned, Figures 5B and 5C were changed to Figure 6A and 6B, and these dates were discussed. (lines 181-192)
In the revised manuscript, we described the following:
We investigated the association of these proteins with GPER using the STRING interaction network. As shown in Figure 6A, ROCK1, TAGLN2, FCHO2, and GPER showed the indirect interaction via epidermal growth factor receptor (EGFR). The correlation between these gene expressions and overall survival (OS) was investigated in the KM plotter (www.kmplot.com). To analyze the data, we used the mRNA sequencing and OS of patients with pancreatic ductal adenocarcinoma (n = 177) and set the best cutoff for comparing their OS with high and low gene expression (Figure 6B). In the case of GPER, we considered that estrogen could regulate the action of GPER, and then compared the correlation between GPER expression and the OS in female and male patients with PC. High GPER expression decreased hazard ratio (HR) and delayed the OS of patients with PC. Interestingly, GPER levels were significantly positively correlated with OS in female patients with PC (p = 0.0086). Low ROCK1, TAGLN2, and FCHO2 expression delayed the OS of patients with PC.
- In reading the discussion, I find that the data as presented do not allow for the conclusions drawn to be made, in particular at lines 214 and 227. Moreover, there is no real evaluation of the data. No alternative hypotheses are presented. Consequently the impact of the study is very limited as presented.
> Thank you for your valuable comment. I agree that the conclusion is limited as you mentioned. We investigated the proteomic analysis and then found tumor tissues significantly differentially expressed several factors (ROCK1, TAGLN2, and FCHO2). Thus, the relation between these factors and GPER through several databases (STRING, Kaplan-Meier plotter) was further investigated. We added the manuscript and modified the conclusion as you mentioned. (lines 240-257).
In the revised manuscript, we described the following:
Particularly, the differentially expressed genes between control and chrysin-treated tumor tissues were investigated and found eight genes (Table 1). The interaction of these genes and GPER were investigated using STRING proteomics. Three factors were identified: ROCK1, TAGLN2, and FCHO2. ROCK1, TAGLN2, FCHO2, and GPER all showed interactions with EGFR in common. Chrysin was reported to inhibit EGFR in breast cancer stem cells [44]. We will further study the correlation of chrysin and EGFR in PC. Remarkably, differentially expressed genes showed that low expression in patients with PC prolonged OS in patients with PC (Figure 6). ROCK1 plays a role in the metastasis of cellular movement, and accumulation of extracellular matrix in cancer-associated fibroblasts (CAF), thus demonstrating important signaling pathways in cancer progression. In PC, ROCK1 is highly expressed, and its inhibition decreased tumor cell growth and CAFs in a previous study [45]. Additionally, TAGLN2 also shows higher expression in PC cells than in normal corresponding cells, as previously reported [46]. Further, knock-out of FCHO2 increased chemosensitivity in PC MIA-PaCa2 cells in a further study [47].
In conclusion, our results suggest that chrysin-induced GPER activation decreases ROCK1, TAGLN2, and FCHO2 expression and subsequently suppresses the proliferation of PC in MIA PaCa-2 cells-derived xenograft model. Therefore, chrysin shows its availability as a chemotherapeutic agent for PC therapy.
I would like to emphasise that although I have been critical of the data and its presentation, it is possible that there is information of significance to the research community here. In the present form, however, I cannot recommend publication of this paper.

Round 2
Reviewer 3 Report
The authors have responded positively to the comments and recommendations in my previous review. Although I would personally preferred having more statistically significant data to publish, it is not my paper. Readers will evaluate the content of the paper according to their interpretation of the data. The revised version is clearer and more balanced. I would like to suggest only one further change which is in the new sentence at the end of the discussion at line 258. I don't think that the results support the use of chrysin itself as an anticancer agent. It would be acceptable, however, to write something like 'Therefore chrysin-like agents could have a role in PC therapy.' I think that this statement is consistent with the results.
Author Response
The authors have responded positively to the comments and recommendations in my previous review. Although I would personally preferred having more statistically significant data to publish, it is not my paper. Readers will evaluate the content of the paper according to their interpretation of the data. The revised version is clearer and more balanced. I would like to suggest only one further change which is in the new sentence at the end of the discussion at line 258. I don't think that the results support the use of chrysin itself as an anticancer agent. It would be acceptable, however, to write something like 'Therefore chrysin-like agents could have a role in PC therapy.' I think that this statement is consistent with the results.
> Thank you for your valuable comment.
As you mentioned, we changed the sentence in the conclusion.